# Aesthetic Diagnosis in Gestalt Therapy [note 1]

**DOI:** 10.3390/bs7040070

**Published:** 2017-10-17

**Authors:** Jan Roubal, Gianni Francesetti, Michela Gecele

**Affiliations:** 1Psychotherapy Training Gestalt Studia, Training in Psychotherapy Integration, Center for Psychotherapy Research in Brno, Masaryk University, 601 77 Brno, Czech Republic; 2International Institute for Gestalt Therapy and Psychopathology, Poiesis. Turin Center for Gestalt Therapy, 10143 Turin, Italy; gianni.francesetti@gmail.com; 3International Institute for Gestalt Therapy and Psychopathology, 10143 Turin, Italy; michelagecele@gmail.com

**Keywords:** psychotherapy diagnostic process, aesthetic diagnosis, Gestalt therapy

## Abstract

The diagnostic process in psychotherapy using the aesthetic evaluation is described in this article. Unlike the classical diagnostic process, which presents a result of comparing clinicians´ observations with a diagnostic system (DSM, ICD, etc.), the aesthetic evaluation is a pre-reflexive, embodied, and preverbal process. A Gestalt Therapy theoretical frame is used to introduce a concept of the aesthetic diagnostic process. During this process, the clinicians use their own here-and-now presence, which takes part in the co-creation of the shared relational field during the therapeutic session. A specific procedure of the aesthetic evaluation is introduced. The clinical work with depressed clients is presented to illustrate this perspective.

## 1. Introduction

Gestalt therapists have often held caution towards psychopathology and diagnostics. It has not been an easy relationship for epistemological, historical, and political reasons [1,2]. Nevertheless, Gestalt therapy has a specific psychopathological understanding: each psychotherapeutic model has one, explicit or implicit. The lesson of humanistic movements—the uniqueness of each person and experience—always remains precious. The intention of developing Gestalt psychopathology is not to label the clients, rather to understand their suffering from an experiential and relational theoretical point of view. Such an understanding provides therapists with a specific support and guidelines for experiential and relational work.

Gestalt therapy understands individual symptoms and human suffering as phenomena emerging from a wider relational field [3] and can offer an original key to understanding, staying with and supporting people who suffer. Moreover, to see human suffering as field phenomena opens up the possibility of understanding the individual in a wider social horizon and the social field in the light of the individual experiences.

In the contact process, human suffering can be met by the therapist who resonates [4,5] with the client and this implies a transformation that is aesthetic. By aesthetic we mean: (1) that it is perceived by senses (*aisthesis*, in Greek, means to perceive throughout the senses), (2) that follows the rules of the figure forming described by the Gestalt psychology [6], and (3) that can be felt as a feeling of something beautiful emerging during the session [7]. The suffering emerges in the therapeutic session as a phenomenon co-created by the client and the therapist and it can be transformed in the process of contact [8]. In this article, we will first introduce briefly the Gestalt therapy approach to psychopathology. Then, we will describe the diagnostic process using the aesthetic evaluation as a pre-reflexive, embodied and preverbal process. Finally, the clinical work with depressed clients will be presented to illustrate this perspective.

## 2. Suffering of Relationship: Gestalt Therapy Approach to Psychopathology

For Gestalt therapy, a continuum exists, without clear-cut distinctions, between healthy and so-called pathological experience. It is on this basis that all attempts at diagnostic categorization and nosology have always been treated with caution [6]. The value given to momentary experience and to the contingency of each and every situation underpins the legitimacy of all lived experiences. This value, common to all of the modalities belonging to the humanistic tradition, helps the therapist to not crystallize people and their suffering’s experiences into fixed *Gestalten*.

Etymologically, the word “psychopathology” consists of three roots: ‘psycho-’, ‘-patho-’, ‘-logy’. *Psyche*, meaning “soul” in Greek, derives from *psychein*: “to breathe”. *Patho*, from the Greek *pathos*: “affection, suffering”, derives from *paschein* (indeurop.): “to suffer”. *Logos*, in Greek: “discourse” [9]. Hence, psychopathology is a discourse on the suffering of the breath, of something elusive, which cannot be confined within a stable object form. It is the suffering of the animating breath, the suffering of the animate living body (in German: *Leib*), not the object-body (in German: *Körper*). All living bodies are living precisely because they have intentional contact with their environment [10]. Psychopathological phenomena concern subjects as they interact with the environment, or more precisely, the interaction of subjects with the environment.

At this point, we come to a radical bifurcation. We can focus on psychopathology as either the suffering of the individual and to consider just this level of exploration, i.e., by focusing the symptoms that the client refers; or we can consider the suffering as an expression in the individual of the suffering of a wider field. There are many and different definitions of field in psychotherapy [11,12,13,14,15,16,17,18,19,20] and in Gestalt therapy [21,22,23,24,25,26,27,28,29,30,31,32,33,34,35,36,37,38,39,40,41]. Here, we refer to a phenomenal field of experience: “by ‘phenomenal field’. We mean that the horizon within which emergent experiential phenomena is generated in the encounter. It can be considered as the horizon of all possible forms, constituting the possibilities (in this field many different forms of experience can emerge), and *limitations* (in this field not all forms of experience can emerge). It is a region of space-time in which a force produces an effect (Maxwell) (Cf. the concept of magnetic field introduced in the 1840s by Maxwell and Faraday: “the region where a particular condition prevails, especially one in which a force or influence is effective regardless of the presence of a material medium” (The New Oxford Dictionary of English, 1998, p. 680)). The phenomenal field is generated by all that is relevant and extends into space and time as far as it can produce a difference in experience—these are its boundaries” [42]. This change of focus opens up two very different universes and two profoundly different ways of approaching psychological suffering.

These two perspectives on the reality of mental suffering can be likened to the two perspectives through which light can be understood in physics: is it a wave or a particle? Reality depends on the way we investigate the world. Psychopathological phenomena are much the same. Psychopathology can be considered a phenomenon belonging to the individual or a phenomenon emerging from the field, belonging to the *Zwischenheit*, to quote Buber [43,44,45,46,47]. In more strictly Gestalt theory terms, it is an emergent phenomenon that happens at the contact boundary (Contact boundary is a theoretical construct of Gestalt therapy: it is the phenomenological boundary where the subject and environment meet, it is a co-created and dynamic ‘third’ space, not reducible neither to the organism nor to the environment, where the experience springs out. See also [48])

Our epistemology is founded on the consideration that experience does not strictly belong only to the subject, nor only to the environment [6,47,48,49] Rather, experience emerges as a “middle voice” at the contact boundary. If we view psychopathology as emerging at the contact boundary, then strictly speaking it is not the subject that suffers. What suffers is the relationship between the subject and the world: that space that the subject experiences, and in which it becomes animate. Psychopathology, in this view, is the pathology of the relationship, of the contact boundary, of the between [50]. The subject is the sensible and creative receptor of this suffering: the subject can feel pain.

From a Gestalt perspective, symptoms are products of a creative self and display human uniqueness [6]. Psychopathology thus represents a unique creative adjustment in a difficult situation. When it becomes fixed, it stops serving the needs of the individual and their environment; it narrows the individual’s spectrum of potentials. The symptoms are viewed not as discrete items, but as a narrowed spectrum of functions [51]. The symptoms indicate limited flexibility in the reactions of the clients. They are then limited in their ability to have fluent contact with their environment. They are not able to act in accordance with their actual need and potentialities, but their behaviour and present experiencing are determined by fixed patterns. They follow a habit, not a deliberated choice [52].

Psychopathological symptoms are phenomenologically observable manifestations of fixed *Gestalten,* which are relational co-creative phenomena of the field. These rigid patterns cause suffering of the contact boundary and of relationships (of course the individual contributes to the organization of their relational field). They become a figure also in the therapeutic relationship: both client and therapist are co-creators of the psychopathology, which emerges in their relationship. Therapists can step out of the rigid field formation using their awareness. In that way, they give support to the relationship and offer to the clients a chance of widening their spectrum of possibilities. The therapists provide a contact experience that was missed by the clients and which they were seeking [45,46,53]. In this sense symptoms are always a plea for a specific relationship: a kind of contact where the symptoms are not needed anymore [54].

What is missed emerges in therapy as a need for a specific and new contact experience. This is the relational need that the client is looking forward to satisfying—or of which to become aware and be recognized—in therapy, it is their interrupted contact intentionality (intentionality for contact is a theoretical construct of Gestalt therapy that comes from the concept of intentionality of the phenomenological tradition [55]. With it, we refer to the emerging tension towards the contact between the therapist and the client, which moves the interactions towards the actualisations of the potentialities in the present therapeutic situation [3,6], it is at the same time their history and their next step). All suffering has its relational “next” towards which it is oriented and which illuminates its meaning [53,56,57]. In giving support, the fundamental question orienting the therapists is “towards which relational experience is the person headed?” The answer to this question marks and points the direction of therapy.

## 3. Gestalt Therapy Approach to Diagnosis

The mistrust of Gestalt therapists towards diagnostics warns us of the risk of becoming experts for the lives of our clients, the risk of treating our image of the client and not meeting the client. However, it is important to realize, that we cannot avoid making some kind of diagnosis. Every experience is random, changeable, amorphous, and chaotic in the moment of its birth [58]. A basic human tendency is to organize each experience into a meaningful structure. We organize our experience of the presence of other people, we give name to our experience, and we give it a structure. We label our surroundings all the time. However, in the position of a therapist we must do it with the client’s benefit in mind and constantly reflect on the process of formulating a diagnosis.

When a therapist meets a client, they encounter an enormous amount of complex information. It comes from various sources: through the therapist’s senses; from their own emotional and bodily experiences; from immediate thoughts and intuitive insights and previous personal and professional experiences that come to mind during the meeting; and, from the theoretical concepts and assumptions that a therapist has assimilated during their education. To process all of this information a therapist needs filters and concepts that help them organize it in a meaningful way. This is necessary for good enough therapy, for contact that is healing and not re-traumatizing, for identifying realistic treatment aims and procedures, and also as a foundation for a responsible creativity on the part of the therapist.

From a Gestalt therapy point of view, diagnosis is a process of naming the emerging meaning of the complex and changeful clinical situation [59]. Gestalt diagnosis is not pointed at fixed conclusions [60] but serves as a flexible and momentary working hypothesis [61], which enables the therapists to orientate themselves in a clinical situation and to consider accurate therapeutic paths. Diagnosis is most useful when kept descriptive, phenomenological, and flexible [62]. The therapists co-create and continuously correct the diagnosis through dialogue with the clients. The therapists who are formulating a diagnosis represent an inseparable part of the actual web of relations and, thus, the phenomena of the interaction between the therapists and the clients are important objects of the therapists’ explorative interest. Diagnosis needs to be able to gauge and communicate the suffering of relationships. What the therapists seek to bring out is the way that a relationship suffers, and which intentionality needs to be supported during contact.

Diagnosis can support the therapeutic relationship by anchoring therapy in an extended corpus of knowledge and experience, in a sedimentary and shared history, in the professional community. The “metaposition” or “other space” that is gradually co-created with the client constitutes a “third” party in which to anchor the therapeutic relationship [63]. It is a space that emerges from the therapists’ need to orient themselves, to read the experience co-created with the clients, and to avoid merging with that experience. It is a space that emerges from the client’s need to believe that there is a starting point and, therefore, an arrival point.

Extrinsic diagnosis can support the therapeutic relationship by anchoring therapy in a “third party” [63]. It emerges from the therapists´ need to orient themselves, to conceptualize the experience co-created with the clients, and to avoid merging with that experience. Extrinsic diagnosis can also help to support contacting where the clients feel the need to express their experience in words and compare them to the words and background knowledge of the therapist. In this case, diagnosis is part of a much broader process of definition and the construction of personal acknowledgement. Finding the words to describe one’s suffering together with the therapist can prove a profoundly meaningful and transforming experience, as it is the result of co-creation within a hermeneutic framework [64]. In this process, both therapist and client bring their understanding of what is happening (included what is described in the diagnostic nosographies) and they try to find a shared way to name and define the client’s experience. Singularly, they do not possess the truth, they are rather looking to co-create, by an embodied dialogical way, a common way to describe the client’s and therapist’s reality. They co-create a shared horizon that supports the therapeutic process.

The Gestalt therapists are grounded in the here and now encounter with the client, they understand the situation in a certain way, orientate themselves in it and accordingly direct their interventions. A metaphor of travelling seems useful here. In psychotherapy, the clients and the therapists set out on a journey of discovery together. The therapists have a specific role and responsibility, sometimes they lead, sometimes they let themselves be led. Together, with the clients they discover the interesting, useful and risky features of the territory. They can travel with or without a clear goal.

They can get lost. The therapists need to stop then and look at maps to get orientation. If this is the case in the clinical situation, the therapists need to withdraw temporarily and let themselves take time so that the therapeutic situation can give a meaning to them. Then they can give a name to this meaning, which is a diagnosis. For the moment, the therapists temporarily and consciously do not focus on the client and the relationship, rather they focus on the description of the meaning of the situation which represents a “third” party there. However, by changing focus, the therapists does not escape from the contact with the clients, rather supports the contact with them, as though pointing out a position on the map and getting directions for a common journey. For example, interventions supporting the therapeutic relationship would be heading in different directions when therapists and clients are part of a borderline field or when they are part of a psychotic field.

## 4. Aesthetic Diagnosis

There are two kinds of diagnosis when orientating towards a therapeutic relationship [1]. The first one, which was briefly described above and may be called *extrinsic* or *map diagnosis.* It results from a comparison between a model of the phenomenon and the phenomenon itself and is created when the therapist consciously focuses on the description of the meaning of the situation. However, when facing the client, the therapists cannot always stop for a moment and consider how they understand the situation. In practice, they can only do this from time to time and maybe mostly after the session. In the live dialogue, the therapists respond immediately. They react by a word, gesture, or tone of voice in the blink of an eye. Also, here, they have guidelines that help them to direct their response. These are guidelines not reached by changing a focus (a temporary switch of a focus from the territory to the map) but on the contrary, by being fully involved in the flow of the relationship. The therapists feel completely involved in the contact process and they act in supporting the relationship as a whole.

The second kind of diagnosis can be called intrinsic or aesthetic diagnosis, which is the specific diagnosis of Gestalt therapy. It arises from the aesthetic criterion (Joe Lay, in [65]) and it is the perception of the aesthetic qualities of what happens, or what fails to happen, that orients the therapists in adjusting their manner of being-with the client. We can compare the extrinsic diagnosis to a map of the territory of the therapeutic situation. The intrinsic diagnosis we can then see as a sense of direction that therapists feel during their journey through the territory. Both kinds of diagnosis serve the therapists for better orientation, but each does so differently. A map provides overview and understanding, a sense of direction is important for immediate decisions and movement in a blind terrain.

“There are two kinds of evaluation, the intrinsic and the comparative. Intrinsic evaluation is present in every ongoing act; it is the end directedness of process, the unfinished situation moving towards the finished, the tension to the orgasm, etc. The standard of evaluation emerges in the act itself, and is, finally, the act itself as a whole” [6] (pp. 65–66). Instant after instant, interactions between the therapist and the client take place unpredictably and chaotically, bringing into play thousands of elements every fraction of a second. Interaction is incredibly complex: it is visual, aural, tactile, muscular, glandular, neurological, gustatory, and olfactive, reactivating layers of memory that fluctuate in waiting, ready to participate in forming a figure. Moreover, it involves expectations and comparisons with thousands of contacts and faces. What orients us in this complexity?

The orientation is enabled by a sensed aesthetic evaluation, which emerges from moment to moment from the contact boundary. It offers orientation for the therapist, it is knowledge (*gnosis*) of the here and now of the relationship through (*dia*) the senses. This act of diagnosis is not a comparison between a model and a phenomenon. We shall call this second kind of diagnosis “*intrinsic or aesthetic diagnosis*”, because it is intrinsic to the process and because it is based on the perception throughout the senses.

The criterion that orients the therapists to support the emerging intentionally for contact is intrinsic to the present moment of the relationship. It is an *aesthetic criterion,* based on the senses and oriented towards a direction. During the contact process, a tension emerges towards the creation of a “good Gestalt”, an experiential figure subjectively perceived as good, according to the rules of the Gestalt psychology [6,65,66,67]. In the flow of the therapeutic encounter, when the natural dynamics of the human meeting evolves from moment-to-moment, the extrinsic evaluation methods, based on a comparison between what happens and an external norm taken as a benchmark [6], are not useful. However, it is helpful in case of case conceptualization, which the therapists need to make between the sessions, or in critical moments of a session, as described below. The therapists perceive continuously the contact qualities and creatively adjusts their presence at the contact boundary: this constitutes the unity of the diagnostic and therapeutic act [65,6]. By sensing the drops of intentionality and losses of spontaneity, the therapists re-position themselves in the relationship, co-creating and supporting it, moment by moment.

This kind of orientation is based on the intuitive evaluation of a contact situation: it is a specific kind of knowledge that emerges at the contact boundary in a moment when the organism and environment are not yet differentiated. For this reason, the aesthetic knowledge is implicit (pre-verbal) and already attuned to the intersubjective dimension [7,68,69]. Guidelines for the next intervention are immediately evaluated according to aesthetic criteria. Only later can the therapists name the process of making their decisions for particular interventions: “It seemed the right thing to keep silent and just to look at the client´s eyes in that moment, it just fitted well”; or “I would not dare to confront the client in that situation, it did not seem appropriate”, etc. Such kind of evaluation is pre-cognitive and pre-verbal and implies not only a passive act, but also an activity, leading the therapists straight to intervening action. Time is not spent then on cognitive processes in the flow of the moment-to-moment interactions.

Working with intrinsic diagnosis, we use intuition as a source of support for therapists. Most immediate interventions are not made from a conscious cognitive deliberateness, but the therapists’ awareness orients them throughout the aesthetic criteria. Often, only after the session can the therapists find a way of describing verbally and understanding cognitively what they did and what were the reasons for the interventions. It does not mean that the therapists work chaotically. Their understanding of the clinical situation and their interventions are led intuitively. Their intuition is cultivated by experience and training. Cultivated intuition enables the therapists to perceive more sensitively slight shades of the therapeutic situation and intervene immediately in an appropriate way even without a cognitive processing. Intuition can lead them in the space “in between” through a soft web of minute signals, for which words and thoughts are too rough instruments. It can be seen as a phenomenological attitude of being present to the other and follow the flow of the co-created experience [70].

What does it really mean making an intrinsic kind of diagnosis? To be aware, awake, with senses active, and at the same time relaxed, allowing you to be touched by what happens [7,71]. To remain confident that chaos does indeed make “sense”, and that with sufficient support a meaning will emerge. The therapists are not disoriented, but present. They are not idle, but are ready to join the “dance” that unfolds at the boundary where clients and therapists make contact. The therapists are ready to gather intentionality and to support its unfolding. It is the intentionality towards contact that brings order to intersubjective chaos. When the arrow of intentionality loses energy and falls, it is recovered by the therapists, who give it new momentum. When the arrow falls and is recovered and re-launched, the emotive intensity of the moment is heightened. Moments of fullness of contact are always unpredictable: we do not know when they will occur, in which minute or second of contacting. They do not occur by chance though: it is the therapists who help deliver those moments by supporting the intentionality of the clients as it unfolds second by second and encounters the therapists’ own intentionality [72].

Intentionality orients the therapeutic process. A loss of momentum, a drop, or interruption in intentionality will prompt the therapists to intervene: intervention may also be silence, immobility, or almost imperceptible movement. The intervention is directed towards the completion of a Gestalt, supports the potential that is ready to appear. How do the therapists notice the movement or interruption of intentionality? The answer lies in being present at the contact boundary, with senses alert and an awareness of one’s bodily, emotive, and cognitive resonances. These resonances emerge indistinctly, not by cognitive process, but rather by giving time to unfold, and only through later reflection can they be distinguished.

A rigorous criterion is what guides this awareness: *the aesthetic criterion* (Joe Lay, in [65]) that leads therapists and clients to co-create a good Gestalt of contact. Again, in this diagnostic approach, no comparison is made between a model of the phenomenon and the phenomenon itself, as happens with diagnostic maps. Here we have the perception of the fluidity of what happens, or what fails to happen, which is what orients the therapists in adjusting their manner of being-with the clients. It is a note out of key, a brushstroke out of place, a touch too much or a touch too little, a little too soon or a little too late. It is not an *a priori* model that guides us, but the unique, special aesthetic qualities of a human relationship in that specific situation. Just as we know how to recognize a note out of key, we can sense that something is out of place or out of time, or so indefinably strange or fatigued in ongoing reciprocal responses. “What happens in a session and which we feel to be “beautiful” is neither objectively beautiful (it is not a quality of the object) nor subjectively beautiful (for me alone, as if it were just a question of personal tastes). It is, indeed, present for whoever is present in their senses—who is, therefore, aware and participating, implicated and resonating in the situation. It is beautiful for us to be present in as much as we are touched by what is happening. We are not, in fact, referring to the beauty of either an object from which we can be detached, nor of something that is “nice”, gracious, comforting, and cosmetic. When involved in contemplating beauty our eyes change, our breath changes: the beauty does not belong to the object or to the subject, but is an emerging contact phenomenon. We are rather concerned with a phenomenon that transforms and seizes us, whose power can have the emotionally disruptive force of a tidal wave or the subtle, penetrating quality of the air high up in the mountains. Moreover, because it transforms, it leaves behind a trace of itself. (...). The link between aesthetics, awareness, the lifeworld and transformation emerges even more clearly if we probe the etymology of the word in further depth, as did the classical philologist Richard Onians:
The Greek verb *aisthanomai* (long form of *aisthomai:* ‘to perceive’), from which *aisthesis* derives, is the middle of the Homeric *aisto*, which means ‘I gasp,’ or ‘breathe in’. In its affinity with terms indicating the “breath” of the living, *aisthesis* shares the same root as *aion*, meaning time which regenerates itself and, prior to that, the “vital force” which flows through bodies (…).[69] (pp. 74–75). [7] (p. 7)

The cardinal points of this “second by second” diagnostic approach is in the here (the experience of space) and now (experience of time) of lived experience, as it manifests itself at the contact boundary. The therapists are the sensitive needle to changes in these seismographs, which record (via individual resonances) the aesthetic values of the relationship here and now, and not individual parameters. The therapists gauge these variations and continuously position themselves in relation to them, with sensorial-physical unity. In the case of a traumatized client, for example, the aesthetic sensitivity “warns” the therapist not to use potentially re-traumatizing too expressive interventions, and it “guides” the therapist rather to focus on the safe here-and-now therapeutic situation. In this way, the therapists do not only bring about the intrinsic diagnostic act, but also the therapeutic act itself: this constitutes the unity of the diagnostic-therapeutic act [6,65]. Sensing the interruption of intentionality, the therapists re-position themselves in the relationship, guiding and curing it, moment by moment.

## 5. Depressing Together: Example of Specific Clinical Aesthetic Evaluation

The gravity of a client’s depression can be measured in terms of their detachment from the in-between, of the degree to which they are absent from the contact boundary. The in-between is the common ground that we constantly co-create at the contact boundary. It is the fabric that connects us to the world and to life moment by moment. In cases of a severe depressive experience, this common ground has ceased to exist and can therefore no longer be traversed. Herein lies the unique quality of melancholic experience. The in-between is no longer a meeting place [73].

It is impossible to co-create a figure of contact. This dysfunction lies at the heart of the therapist’s difficulty in connecting with the client, in ensuring the usual comings and goings of resonances, consonances, and dissonances, which should fill up therapeutic space and time. In short, nothing reverberates in the therapeutic in-between, which is aesthetically perceived by the therapist and thus constitutes the intrinsic diagnosis in the contact with the depressed client.

A central facet of depressive experiences is the lack of any interest. This does not simply mean that the subject is not attracted to or involved in anything. It also has the more radical implication that they are no longer in the “*inter*” of “*esse*”, which they in some sense are removed from being in the in-between itself, from the nerve centre where all the infinite strands of life knit together [74]. The sense of lifelessness, which is perhaps one of the most distinguishing features of depression, is clearly a manifestation of this condition. The healthy growth of the self requires that the organism be at once separated from and welded to the world. This connection with the world is what is lacking in the severe depressive experience.

Severe depressive experiences are characterized first and foremost by a certain sluggishness in the figure/ground dynamics: the figure strains to emerge from a ground which is devoid of energy. There are neither interests, stimuli nor impulses of intentionality. The client often remains silent and immobile on the chair throughout the session. Not even the vaguest hint of a figure peeks through. Nothing is relevant. Nothing means anything, since meaning itself is developed at the contact boundary in the figure/ground dynamic, where the figure acquires size, depth, and meaning through its relation to the ground.

No intentionality emerges, since intentionality does not belong to any one individual but rather emerges and reveals itself through contact: it is the force that drives all of our encounters at the contact boundary. When we enter into a severe depressive relational field, our senses encounter a nothingness, a torpid wasteland that seems at some times to be made of stone and at others of a fluid, all-engulfing fog. “My head’s full of a kind of fog, which shifts continuously without ever taking on any distinct form. I’m really confused. I don’t know what to do”. At other times again, it seems as if nothing has any meaning: “I look out at the view as if it were nothing more than static pictures on a flat screen. The mountains, which have always been a source of joy to me, are now just there: unreachable, inert and useless. Nothing appeals to me. There’s nothing I can relate to, nothing that means anything to me”.

The therapists perceive the lack of direction in a dilation of time and space. These two transcendental cornerstones of human experience have been altered. It would be inaccurate to say that the figure makes use of space and time as pre-existent categories. Rather, time and space emerge at the very moment at which the figure is co-created in the present. When depression creeps up, the present moment fails to emerge. It lacks the support of both the previous moment that is coming to an end (*retentio*) and of the subsequent moment, which is coming into being (*protentio*).

When the therapists situate themselves in the relational field of the client, they will become immediately aware of this modified sense of time, which has been dilated to the point of suspension, to a point at which it has almost come to a complete stop. Space, meanwhile, is in a state of constant expansion. The distance between the therapist’s chair and that of the client seems ever greater, to the extent that it comes to appear insurmountable. The energy required to traverse it comes to appear impossible. However, the very fact that its apparent absence causes such acute distress demonstrates that intentionality is actually present. It is present in the very pain, which derives from the perception of its absence. If the painfully felt absence of intentionality exists in the figure, then intentionality must be present in the ground.

The depressive experience is situated within a relational field. Time and space are the roads that we conceive of ourselves as we make our way towards that which is loved and necessary. They are relational-dependent variables, generated through the impetus of the journey itself, which is never just a single movement but always a co-movement. When this movement fails, what we experience is the abyss, which separates us. The affective bridge, upon which our very selves are constituted and from which subjectivity springs, is lost. Depressive experiences are the expression in the individual of a specific relational experience: namely the impossibility of reaching the other. Depression is the way in which the subject experiences the surrendering of hope in the face of the ineffectiveness of their vain attempts to reach the other. Depression can be understood as a co-constructed relational phenomenon with three intrinsic and essential features: a profound attachment, whereby the other is loved and necessary, the failure of all efforts to reach the other and the emotive absence of the other from the relationship.

The depressive experience of each person is a unique one; it is always an inseparable part of the unique person’s life story. The depressive experience also has an interpersonal nature, it is a co-created phenomenon: it appears in relationships and there it is maintained. Considering the context of the life story and web of relationships, the depressive experience can be seen as a function of the field, as a form of creative adjustment. It can help a person to survive a difficult situation, it can signal a life transition and re-focus the search for life meaning, it can facilitate a change in frozen habitual relationship patterns, etc. However, if a person uses a depressive way of relating in a rigid and stereotypical way in their life, the depressive functioning becomes a *fixed Gestalt*. It can be described as a vicious circle, which decreases the ability of the organism to cope with its own mental and physical processes as well as external demands. It leads to more frequent failures, subsequent deepening of the depressive state, and a further decrease in the capacity of the organism. The originally useful adaptive mechanism of depressive adjustment [75] may turn into an exhausting and devastating fixed Gestalt of depression as a kind of suffering that significantly limits a person’s capacity to creatively adjust. Conceptualizing depressive experience as a fixed Gestalt of the originally useful way of creative adjustment can serve as an extrinsic diagnosis, which provides the therapists with an orientation in the complexity of the depressed clients´ relational history.

The usual organization of the relational field described above tends to be repeated in the therapeutic situation too. The therapist becomes a part of the “depressive organization” of the field. The usual reaction of the client’s family or other nearest and dearest persons to their depressive state is polar. They first want to encourage her/him (“Come on, it will be OK soon. Let’s have some fun, it will help you to overcome this”). Later, when this effort is not effective and they become exhausted, they try to protect themselves and withdraw from the depressed person (often with more or less hidden aggression).

The therapists find themselves in the same relation pattern and they feel impulses to repeat the described reactions to the depressive person. The therapist can, for example, try to encourage the client with well-intended practical suggestions for changing stressful life conditions, and then experiences frustration or irritation, when the client is not able to make any change. Thanks to their awareness, the therapists have the chance to step out of this rigid relational pattern and respond differently to the depressed person—they remain available for contact, do not blame either themselves or the client, do not give up hope. In doing this, the therapists change the usual rigid field organization and opens a space for a change also for their client.

Fear is a common initial reaction when dealing with a severely depressed client. This may take the form of an undefined yet powerful sense of unease or of an intense fear for the client. Sometimes the therapists may wish to get away from the client, or to send them on to be dealt with by someone else. It is important to frame these experiences in their field context. All of these reactions reflect the therapists’ perception of the lack of ground in the relational field. It is for this reason that the involvement of a third party provides a vital anchor [1]. This may take the form of pharmacological support, supervision, meetings with colleagues, or further theoretical training (hopefully including reading this chapter).

Another aspect of counter-transference concerns the side effects of the therapists’ placing themselves in a depressive field. The depressive condition leaves the therapists teetering on the edge of a precipice, feeling a terrible weight pulling them down towards into the abyss, the vacuum, a state of solitude, fear, and extreme impotence where all sense of direction is lacking. This can lead to feelings of anger, which may result in self-depreciation (“I’m not up to working with this client”) or a loss of faith in one’s training and profession (“My chosen therapeutic approach doesn’t equip me to deal with this client” or “Psychotherapy’s no use at all with these clients: they just need medication!”).

The therapists’ experiences with a depressive client can be described by an overall metaphor of “magnetic power of depression”. The therapists feel drawn to the client experience as to a magnet. They then either keep a safe emotional distance by keeping a professional mask, keeping the depressive experience unfamiliar for themselves and sometimes taking an inappropriate responsibility for the whole situation. Or, the therapists come closer by sharing the client’s depressive experience to some extent. The therapists experience falling off, loneliness, helplessness, shame, and heaviness. In this case, they might feel endangered by the risk of “depressive contagion”, they experience: “It is too much for me!” and react by self-protection and/or aggression towards the client: “She’s unbearable. She needs me and I’m here, holding out my hand but she just can’t see it!” or “Nothing I do is of any use, so she can do as she likes and that’s that!”. The therapists may feel tempted to defy or challenge the client: “Ok then, let’s see what’s stronger: my commitment or your inertia!”.

It is important for the therapists to be aware of their own experience and not to blame their client or themselves for it, because blaming is a distinct feature of a depressive field organization. The therapists can use the metaphor of “magnetic power of depression”, and their experience would indicate how strong the “magnetic power” of depression is and what is the therapists’ position towards it.

The therapists themselves are endangered in the depressive field organization. They might get “infected” by the client’s depression and get depressed too. There is a clinically observed phenomenon of spreading emotions associated with depression in interpersonal contacts. The “contagiousness of depression” is a theoretical concept that serves as a tool for better understanding and not for blaming the “carrier” of the depression. This concept has been substantiated by meta-analysis of 40 research projects [76], which gives sufficient support to the statement: “depressive symptoms are contagious in close relationships”. The specific sequence of therapists’ experiences in a depressive field has been explored and described by [77] as a “depressive co-experiencing trajectory”.

The therapists’ task is to remain present, when it would be so easy to get lost, fall asleep, or lose one’s temper, without getting depressed, when it is so easy to lose hope. Such a situation represents one of the most arduous tasks faced by the psychotherapist: they place their own self at the client’s disposition, but in this situation, they may experience as if in the in-between there was an abyss [78]. How can they inhabit such a cavity, such an abyss?

All of the therapists’ experiences should be brought into awareness, because they represent a way of being-with the other in the relational field. The field perspective provides support to the therapists on two counts: it enables them to make sense of their emotions at the same time as enabling them to act. By asking themselves “how we are co-creating the depression here and now”, or “how we are depressing together?” [75] the therapists bring the situation back into range.

## 6. Conclusions

As psychotherapists, we need both the map (an extrinsic diagnosis) and the sense of direction (an intrinsic diagnosis). The extrinsic diagnosis is a basis for the work of a psychotherapist. Whenever we create an extrinsic diagnosis, we are fixing the particular way the field of the therapeutic situation has organized itself. We focus on the description of the meaning of the present therapeutic situation and we do not focus on being with the client for the moment. However, if we burdened ourselves with the demand that we should focus on the flow of the therapeutic relationship all of the time, we would paradoxically limit our therapeutic flexibility. A fluent and nourishing flow of contact can develop if we also allow ourselves time to find orientation and meaning, to anchor in a third party, to diagnose.

We can have several kinds of maps, each describing the clinical situation from a different perspective. We can have a map based on observation of the process of co-creation here and now, another one based on observation of roles and interactions within a system and another one based on phenomenological observation of the symptoms. During the process of psychotherapy, we naturally develop maps to give meaning to our experience. We cannot avoid making some kind of a diagnosis. All that we can do is to remain aware of the process of diagnosing and bring our awareness back into contact with the client. We must keep in mind that a diagnosis is not a description of the person in front of us, it is merely a tool that enables us to meaningfully organize our experience with this person and so it helps us to be grounded and present for an encounter.

The extrinsic diagnosis becomes progressively less important as the therapist gains greater expertise. All travellers need maps to orient themselves, but it is also true that the more experienced a traveller you are, the more you can rely on your sense of direction. Sense of direction is something developed moment by moment during your journey, without the use of too many maps. The intrinsic or aesthetic diagnosis is essential in orienting ourselves moment by moment through interaction. It is fundamental in providing specific support in Gestalt therapy. No map will ever be detailed enough to warn us of the potholes in the road and the bends along the track. No map is ever updated to the point of what is happening here and now. This kind of orientation is sufficient when, after having travelled widely and studied countless maps, the traveller is confident of how to move across unknown territories.

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
