# Peer review of "Aesthetic Diagnosis in Gestalt Therapy"

_behavsci, 2017, doi:10.3390/bs7040070_

Round 1

Reviewer 1 Report

Topic of the paper is timely and of interest to potential readers. The Gestalt therapy perspective on the process of diagnosing is well-presented and carefully explained. References are sound and relevant.

The following are some suggestions for the final version:

p. 3, lines 103-112 would benefit from a more precise definition of what is healthy or not. For example, if a patient wants to intentionally and spontaneously harm himself or someone else, is this healthy? 

line 106 Here the authors state that there is no need to use extrinsic evaluation methods, but later this is refuted and they claim that extrinsic methods can be helpful as maps for orientation. Find a way to correct this seeming contradiction.

A minor re-arrangement for better comprehension: Section "Aesthetic diagnosis" could be placed immediately following line 102; lines 103 - 112 could be inserted after line 194. And the transitions between these paragraphs could then be adjusted accordingly, so that the reader better comprehends what the authors mean by "aesthetic criterion" and also that intrinsic as well as extrinsic diagnosis can be helpful.

p. 6 line 236 Either put the word "dance" in quotation marks or use another word, for example, interaction. Non-Gestalt-therapists might otherwise think that the therapist and patient are really dancing!

p. 6f. Less description of depressive processes and more case examples would be illuminating here, for example after p. 8, line 355.

p. 9 This reviewer had the image of a very heavy weight rather than a strong magnet, but this is a matter of opinion.

p. 9 line 397 "id function" must either be defined (not the same as the id in psychoanalysis!) or circumscribed.

p. 9 line 407 "basis" might be substituted for the word "ground" in this context.

Author Response

Author's Notes to Reviewer 1

Dear reviewer, thank you for your feedback. It helped us to improve our article.

p. 3, lines 103-112 would benefit from a more precise definition of what is healthy or not. For example, if a patient wants to intentionally and spontaneously harm himself or someone else, is this healthy?  I

We agree. Using the term „healthy“ would open a wider discussion. We rather keep the text focused, so we have skiped it.

line 106 Here the authors state that there is no need to use extrinsic evaluation methods, but later this is refuted and they claim that extrinsic methods can be helpful as maps for orientation. Find a way to correct this seeming contradiction. 

The text was amended according to your suggestion. It was specified when the use of extrinsic method is useful (case conceptualization) and when not (the flow of the therapeutic encounter, when the natural dynamics of the human meeting evolves from moment-to-moment).

A minor re-arrangement for better comprehension: Section "Aesthetic diagnosis" could be placed immediately following line 102; lines 103 - 112 could be inserted after line 194. And the transitions between these paragraphs could then be adjusted accordingly, so that the reader better comprehends what the authors mean by "aesthetic criterion" and also that intrinsic as well as extrinsic diagnosis can be helpful.

The lines 103 - 112 were inserted after line 194.

p. 6 line 236 Either put the word "dance" in quotation marks or use another word, for example, interaction. Non-Gestalt-therapists might otherwise think that the therapist and patient are really dancing!

The word "dance" was put in quotation marks.

p. 6f. Less description of depressive processes and more case examples would be illuminating here, for example after p. 8, line 355. 

An example of the therapist part in the depressive relational pattern was added.

p. 9 This reviewer had the image of a very heavy weight rather than a strong magnet, but this is a matter of opinion.

Yes, that is another kind of a metaphorical description, which we also use elsewhere, here we prefer to stay with this one metaphore of a magnetic power.

p. 9 line 397 "id function" must either be defined (not the same as the id in psychoanalysis!) or circumscribed.

A phenomenological description without the theoretical construct of the ‚id function‘ is used.

p. 9 line 407 "basis" might be substituted for the word "ground" in this context.

The word "basis" was substituted for the word "ground".

Reviewer 2 Report

Textual editing would strengthen this work, in particular it reads at times as a series of fragmented assertions, without an overall flow or consistency of voice. Greater clarity, and definition of terms, would strengthen the argument. Many assertions need to be supported by reference to relevant recent academic literature. 

Particular high points are the discussion on the derivation and possible meanings of the term"psychopathology", and the discussion on depression that artfully conveys this as a subjective experience and field phenomenon.

To bring this paper to its full potential as an engaging and important contribution to the discourse on diagnose, I advise attention the following details:

Greater clarity, and definition of terms, would strengthen the argument. There are some run-on sentences, typos (227: eg lead = led, etc) and odd punctuation to be revised.

There is a need to expand clipped or unclear sentences. For example: "Gestalt psychopathology is an understanding of human suffering through our theory, not a way of labelling up clients" (26-7). This does not add to clarity of the meaning. 

A suggestion about the use of pronoun throughout: the singular 'they' is preferable to the use of s/he; and 'their' is preferable to the use of his/her. The trend is to move away from gender binary identifications, and certainly away from generic masculine pronouns as in 84-86.

Avoid interchange of the terms client and patient.

The introduction needs editing to increase precision. Statements need to be supported by referencing; and key terms defined, including  'field phenomenon', which has other resonances in different disciplinary domains. And in the 3rd paragraph the discussion about the 'aesthetic' need to be clear and referenced.

Appeals to "all of us" and "we believe" may have made good sense in the earlier iteration the paper as, in part, a conference presentation.  But in written form it has the effect of reducing the strength of the argument.

Statements in the first paragraph of the section "Suffering of Relationship" are circular (e.g. "the value given to momentary experience ... underpins the legitimacy and the value of all lived experiences", 45-7) or unsubstantiated (e.g. "It is this value that prevents the crystallization into fixed Gestalten of people and their experiences",48).

The third paragraph in this section (58 to 62) is unclear, and requires expansion and more careful punctuation; especially as this section is setting up the argument.

68-9 The term "contact boundary" needs to be defined and referenced

70 - 77 The slippage in the distinct meanings of 'subject' and 'organism' in this section confuses the point. Here the terms and argument of the paper are still being developed, so clarity is critical for the purposes of coherence and credibility.

Again at 78 - 80 there is some confusion in these two sentences:  one asserts that symptoms are products of the creative self and display human uniqueness; the other considers them a co-creative phenomenon of the field. This can be better expressed to greater effect.

Such examples are found throughout the text.

In the discussion on the aesthetic criterion more could be said about how "grace, brightness, rhythm and harmony might manifest - what might they look and feel like? Aesthetic beauty is so inherently subjective, are there some boundaries to this domain that enable therapists to learn and develop? 

Discussion on the "metaposition" or "other space" is promising but not clearly enough expressed. It is described as belonging to the professional community, emerging from the therapist's need and also emerging from the patient's need. Is that what is meant?

Define “hermeneutic framework” for your purposes.

The sentence at 154-5 is unclear, in particular, “…the kind of extrinsic diagnosis used.” Used where and by whom?

The section on getting lost in therapy 162-73 is engaging and promising but, again, suffers a little from a feeling of randomness rather than a flowing consistency of voice. 

[At 200 -203 the description of interaction is wonderful.]

At 204-6 and 209-12 there is repetition of points in the previous paragraphs. Again the structure of argumentation here would benefit from an edit for flow, consistency and for building, rather than restating, the argument.

In 213 -25 the discussion on aesthetic criteria is interesting but not yet set out clearly. The description of the process risks it being merely pre-reflective reactivity; in other words, instinctive. The next paragraph elaborates on this well. Though it would be helpful to have examples of how Gestalt trains therapists for this.

At 237 the breath returns.  If this is a critical theme in the paper it needs to be included in the discussion more fully throughout; especially given it is the basis of your reinterpretation of psychopathology.

The specific use of the term “intentionality’ here, requires definition or contextualization.

245-51 the description here is of the phenomenological process. It would be effective to name is as such, and reference it. 

The summary paragraph of this section 262 -70 brings some clarity and starts to solidify the argument. Again, though, it would benefit from some clear examples of how this works.

 In the section on “depressing together” the descriptions of depression are clear and compelling; in particular what occurs at the cardinal points of space and time; the role of the affective bridge in the constitution of subjectivity; and the  shift from the depressive experience as a creative adjustment to a fixed gestalt. 

At 397 the “id function” appears out of context. Ego psychology has hitherto not been mentioned:  I suggest another term or contextualization of this one.

Overall the argument improves toward the end of the paper, and some of the main points become clearer.   But with a careful revision it could be an excellent publication. 

Author Response

Author's Notes to Reviewer 2

Dear reviewer, thank you for your feedback. It helped us to improve our article.

Textual editing would strengthen this work, in particular it reads at times as a series of fragmented assertions, without an overall flow or consistency of voice. Greater clarity, and definition of terms, would strengthen the argument. Many assertions need to be supported by reference to relevant recent academic literature. 

Particular high points are the discussion on the derivation and possible meanings of the term"psychopathology", and the discussion on depression that artfully conveys this as a subjective experience and field phenomenon.

To bring this paper to its full potential as an engaging and important contribution to the discourse on diagnose, I advise attention the following details:

Greater clarity, and definition of terms, would strengthen the argument. There are some run-on sentences, typos (227: eg lead = led, etc) and odd punctuation to be revised.

We have checked the text and corrected the typos we found.

There is a need to expand clipped or unclear sentences. For example: "Gestalt psychopathology is an understanding of human suffering through our theory, not a way of labelling up clients" (26-7). This does not add to clarity of the meaning. 

The sentence was amended to describe more clearly, what we had in mind.

A suggestion about the use of pronoun throughout: the singular 'they' is preferable to the use of s/he; and 'their' is preferable to the use of his/her. The trend is to move away from gender binary identifications, and certainly away from generic masculine pronouns as in 84-86.

We have amended the use of pronoun according to your suggestion.

Avoid interchange of the terms client and patient.

We use the term „client“ consistently in the whole document.

The introduction needs editing to increase precision. Statements need to be supported by referencing; and key terms defined, including  'field phenomenon', which has other resonances in different disciplinary domains. And in the 3rd paragraph the discussion about the 'aesthetic' need to be clear and referenced.

We agree. The Introduction was modified according to your suggestions.

Appeals to "all of us" and "we believe" may have made good sense in the earlier iteration the paper as, in part, a conference presentation.  But in written form it has the effect of reducing the strength of the argument.

We have checked the document and corrected such expressions where we have found them.

Statements in the first paragraph of the section "Suffering of Relationship" are circular (e.g. "the value given to momentary experience ... underpins the legitimacy and the value of all lived experiences", 45-7) or unsubstantiated (e.g. "It is this value that prevents the crystallization into fixed Gestalten of people and their experiences",48).

We have amended the text according to your suggestion.

The third paragraph in this section (58 to 62) is unclear, and requires expansion and more careful punctuation; especially as this section is setting up the argument.

We have amended this paragraph according to your suggestion.

68-9 The term "contact boundary" needs to be defined and referenced

We have added the definition and a reference.

70 - 77 The slippage in the distinct meanings of 'subject' and 'organism' in this section confuses the point. Here the terms and argument of the paper are still being developed, so clarity is critical for the purposes of coherence and credibility.

Only the word „subject“ is used here now.

Again at 78 - 80 there is some confusion in these two sentences:  one asserts that symptoms are products of the creative self and display human uniqueness; the other considers them a co-creative phenomenon of the field. This can be better expressed to greater effect.

The idea of a symptom as „co-creative phenomenon of the field” was moved to the following paragraph, which describe the relational nature of symptoms.

Such examples are found throughout the text.

In the discussion on the aesthetic criterion more could be said about how "grace, brightness, rhythm and harmony might manifest - what might they look and feel like? Aesthetic beauty is so inherently subjective, are there some boundaries to this domain that enable therapists to learn and develop? 

This we consider this for a very important point. The aesthetic criterium is based on the rules of perception coming from the Gestalt psychology. We have some formal rules for the process of figure forming: we are not neutral, we have preferences, we have a prereflective evaluation of what is happening in our senses. The evaluation is both subjective and objective, it is emerging from the interaction between the organism and the enviroment. We added a paragraph about this

Discussion on the "metaposition" or "other space" is promising but not clearly enough expressed. It is described as belonging to the professional community, emerging from the therapist's need and also emerging from the patient's need. Is that what is meant?

The paragraph was amended in orded to become simplier, easier to understand and without introducing too many new terms.

Define “hermeneutic framework” for your purposes.

The term was defined.

The sentence at 154-5 is unclear, in particular, “…the kind of extrinsic diagnosis used.” Used where and by whom?

The sentence was deleted as redundant.

The section on getting lost in therapy 162-73 is engaging and promising but, again, suffers a little from a feeling of randomness rather than a flowing consistency of voice. 

The paragraph was amended to be more consistent.

[At 200 -203 the description of interaction is wonderful.]

At 204-6 and 209-12 there is repetition of points in the previous paragraphs. Again the structure of argumentation here would benefit from an edit for flow, consistency and for building, rather than restating, the argument.

The paragraph was shortened and partly refurmulated to be more consistent and not restating.

In 213 -25 the discussion on aesthetic criteria is interesting but not yet set out clearly. The description of the process risks it being merely pre-reflective reactivity; in other words, instinctive. The next paragraph elaborates on this well. Though it would be helpful to have examples of how Gestalt trains therapists for this.

The paragraph was reformulated to be more consistent and the examples were amended to be more concrete.

At 237 the breath returns.  If this is a critical theme in the paper it needs to be included in the discussion more fully throughout; especially given it is the basis of your reinterpretation of psychopathology.

We would like to delete it here, since we wouldn’t like to expand to much this issue.

The specific use of the term “intentionality’ here, requires definition or contextualization.

See footnote 4

245-51 the description here is of the phenomenological process. It would be effective to name is as such, and reference it. 

The text was amended according to your suggestion.

The summary paragraph of this section 262 -70 brings some clarity and starts to solidify the argument. Again, though, it would benefit from some clear examples of how this works.

An example was added here.

 In the section on “depressing together” the descriptions of depression are clear and compelling; in particular what occurs at the cardinal points of space and time; the role of the affective bridge in the constitution of subjectivity; and the  shift from the depressive experience as a creative adjustment to a fixed gestalt. 

At 397 the “id function” appears out of context. Ego psychology has hitherto not been mentioned:  I suggest another term or contextualization of this one.

We amended the text according to your suggestion.

Overall the argument improves toward the end of the paper, and some of the main points become clearer.   But with a careful revision it could be an excellent publication. 

Reviewer 3 Report

 The writing is intellectually stimulating, thought provoking and useful for a wide readership, to increase understanding of Gestalt’s philosophy and practice of co-created psychopathology and diagnosis.

The article addresses diagnostic process in Gestalt psychotherapy drawing attention to psychopathology as a co-created phenomenon of the field. This Gestalt perspective is framed against a background of diagnostic methods which consider psychopathology as fixed symptoms, a consequence of the suffering individual. The writing offers a passionate and thorough explanation of Gestalt’s alternate emphasis on psychopathology and diagnosis as co-created at the contact boundary between the client and her/his environment.

Having defined Gestalt process of diagnosis as non-fixable and fluent, the authors go on to identify two different forms of diagnosis, extrinsic and intrinsic. Extrinsic usually happens in reflection outside of the session, while intrinsic is a moment-to-moment intuitive process of being with a client, making sense out of a shared field of dialogue.

Therapeutic work with depression is discussed as an illustration of a diagnostic process. The authors demonstrate their extensive experience of working with depressing within a clinical frame.   The description of ‘depressing together’ resonates strongly in its attention to the embodied sense of being with a client. As with Jan Roubel’s earlier article in British Gestalt Journal (2007), the authors’ reflection on depressing is supportive for therapists and clients.

I offer a minor critique regarding the writing on depression, which is offered as an example of  ‘specific clinical aesthetic evaluation’. As I read, this section feels as if it has been inserted from a different paper, rather than integrated into the discussion. The authors might consider making a stronger link to sections before and after, detailing more transparently how the work with depression offers an example of intrinsic and extrinsic diagnosis.

The issue that I find more problematic in the article is the reference to aesthetic criterion. I understand that the argument I am about to put forward here is influenced by my own cultural background as a white Western therapist/academic and is open to discussion. The etymology of ‘aesthetic’ derives from the Greek notion of sense perception – and if this was all there was to the definition it would be a suitable term to apply to Gestalt processes of diagnosis. However the term aesthetic in Western culture has come to be associated with art and philosophy, therefore with concepts of beauty that are Western, Christian, culturally and performatively acquired - i.e. steeped in Western moral judgments of taste and proper-ness. The authors become party to this culturally defining history as they link aesthetics with beauty, grace, brightness, rhythm and harmony. I would like to suggest that we are in dangerous territory as Gestalt therapists when we refer to aesthetic evaluation as pre-reflexive and preverbal, provoking tensions between what is nature and nurture, internal and external, performativity and essentialism. To use the terms beauty and harmony in the context of therapeutic diagnosis provokes a universalism that is unaware of its own cultural weight. For many clients, resistance to, and rejection of, Western concepts of beauty and grace is exactly where they need to be for their own health. Often, attempting to maintain, and failing to meet, Western notions of beauty and grace is fundamental to a client’s suffering (shame).

Drawing on these terms seems to contradict the authors’ clear definitions in the writing regarding the Gestalt use of diagnosis as a chaotic, uncertain untraveled terrain that the therapist and client embark on together. Furthermore to associate aesthetic criterion to the notion of ‘health’ (line 105) suggests that there is an ideal state of health that we must work towards. As long as Western Gestalt therapists apply a notion of aesthetic of beauty in relation to health, the further we get from understanding the difference and otherness of our meetings with people from diverse cultures. How we bring our cultural history into play in our definitions of health is of paramount importance when working with intrinsic, sensed, intuitive diagnostic process.

Finally, while the authors maintain a focus on aesthetics in relation to beauty and grace they  present a contradiction. On the one hand they emphasize a fluid, unfixed diagnostic process happening in the between-ness of the contact boundary. On the other hand the authors appear to be advocating for Western defined criterion for qualities of health that color the meeting and which carry an (unspoken) binary in their composition (beauty/ugliness, harmony/discord).

I would like to offer two suggestions to remedy this problematic use of language. I am not asking the authors to follow either of these suggestions, as they may prefer to leave the article as it is because of its provocative stance on aesthetics. Firstly, the authors might more carefully frame and contextualize their use of aesthetic criterion from their subjective cultural perspectives. Secondly the authors might consider editing the references altogether. Regarding this second option, I have re-read the article removing the terms aesthetic, beauty, grace, brightness and harmony. The article reads well without and, from my perspective as a reviewer, sustains openness to uncertainty and the unknown potential of co-created contact - which is clearly what the authors are seeking. The writing on aesthetic criterion (lines 103-108) could be edited without losing the core power of the diagnostic theme. Similarly on line 37 the word beauty could be omitted and on line 189 the term grace could be omitted. The term aesthetic in the title could be edited from the title.

The article is definitely publishable on the strength of the discussion about psychopathology, diagnosis and a Gestalt perspective on working with depression. This writing is articulate, accessible and useful for its intelligent, in depth research into the phenomenological process of a therapeutic meeting.

Author Response

Author's Notes to Reviewer 3

Dear reviewer, thank you for your feedback. It helped us to improve our article.

The writing is intellectually stimulating, thought provoking and useful for a wide readership, to increase understanding of Gestalt’s philosophy and practice of co-created psychopathology and diagnosis.

The article addresses diagnostic process in Gestalt psychotherapy drawing attention to psychopathology as a co-created phenomenon of the field. This Gestalt perspective is framed against a background of diagnostic methods which consider psychopathology as fixed symptoms, a consequence of the suffering individual. The writing offers a passionate and thorough explanation of Gestalt’s alternate emphasis on psychopathology and diagnosis as co-created at the contact boundary between the client and her/his environment.

Having defined Gestalt process of diagnosis as non-fixable and fluent, the authors go on to identify two different forms of diagnosis, extrinsic and intrinsic. Extrinsic usually happens in reflection outside of the session, while intrinsic is a moment-to-moment intuitive process of being with a client, making sense out of a shared field of dialogue.

Therapeutic work with depression is discussed as an illustration of a diagnostic process. The authors demonstrate their extensive experience of working with depressing within a clinical frame.   The description of ‘depressing together’ resonates strongly in its attention to the embodied sense of being with a client. As with Jan Roubel’s earlier article in British Gestalt Journal (2007), the authors’ reflection on depressing is supportive for therapists and clients.

 I offer a minor critique regarding the writing on depression, which is offered as an example of  ‘specific clinical aesthetic evaluation’. As I read, this section feels as if it has been inserted from a different paper, rather than integrated into the discussion. The authors might consider making a stronger link to sections before and after, detailing more transparently how the work with depression offers an example of intrinsic and extrinsic diagnosis.

This sentence was added into the second paragraph of the part describing depression to show more explicitelly the connection with the intrinsic aesthetic diagnosis: „In short, nothing reverberates in the therapeutic in-between, which is aesthetically perceived by the therapist and thus constitutes the intrinsic diagnosis in the contact with the depressed client.”

 This sentence was added into the ninth paragraph of the part describing depression to show more explicitelly the connection with the intrinsic aesthetic diagnosis: „Conceptualizing depressive experience as a fixed Gestalt of the originally useful way of creative adjustment can serve as an extrinsic diagnosis, which provides the therapist an orientation in the complexity of the depressed client´s relational history.”

The issue that I find more problematic in the article is the reference to aesthetic criterion. I understand that the argument I am about to put forward here is influenced by my own cultural background as a white Western therapist/academic and is open to discussion. The etymology of ‘aesthetic’ derives from the Greek notion of sense perception – and if this was all there was to the definition it would be a suitable term to apply to Gestalt processes of diagnosis. However the term aesthetic in Western culture has come to be associated with art and philosophy, therefore with concepts of beauty that are Western, Christian, culturally and performatively acquired - i.e. steeped in Western moral judgments of taste and proper-ness. The authors become party to this culturally defining history as they link aesthetics with beauty, grace, brightness, rhythm and harmony. I would like to suggest that we are in dangerous territory as Gestalt therapists when we refer to aesthetic evaluation as pre-reflexive and preverbal, provoking tensions between what is nature and nurture, internal and external, performativity and essentialism. To use the terms beauty and harmony in the context of therapeutic diagnosis provokes a universalism that is unaware of its own cultural weight. For many clients, resistance to, and rejection of, Western concepts of beauty and grace is exactly where they need to be for their own health. Often, attempting to maintain, and failing to meet, Western notions of beauty and grace is fundamental to a client’s suffering (shame). 

We would agree on this critique, if we would consider beauty as a universal and shared concept. However, our point of view is different. We consider the feeling of something beautiful appearing as an emergent and cocreated phenomenon. At the same time, we agree it is inherently cultural dependent.  

Drawing on these terms seems to contradict the authors’ clear definitions in the writing regarding the Gestalt use of diagnosis as a chaotic, uncertain untraveled terrain that the therapist and client embark on together. Furthermore to associate aesthetic criterion to the notion of ‘health’ (line 105) suggests that there is an ideal state of health that we must work towards. As long as Western Gestalt therapists apply a notion of aesthetic of beauty in relation to health, the further we get from understanding the difference and otherness of our meetings with people from diverse cultures. How we bring our cultural history into play in our definitions of health is of paramount importance when working with intrinsic, sensed, intuitive diagnostic process.

We agree. We rather have skipped the concept of health from the paper. It would be interesting to open a wider discussion on this topic, but this article is focused on a different topic.

Finally, while the authors maintain a focus on aesthetics in relation to beauty and grace they  present a contradiction. On the one hand they emphasize a fluid, unfixed diagnostic process happening in the between-ness of the contact boundary. On the other hand the authors appear to be advocating for Western defined criterion for qualities of health that color the meeting and which carry an (unspoken) binary in their composition (beauty/ugliness, harmony/discord).

Please see our answer above.

I would like to offer two suggestions to remedy this problematic use of language. I am not asking the authors to follow either of these suggestions, as they may prefer to leave the article as it is because of its provocative stance on aesthetics. Firstly, the authors might more carefully frame and contextualize their use of aesthetic criterion from their subjective cultural perspectives. Secondly the authors might consider editing the references altogether. Regarding this second option, I have re-read the article removing the terms aesthetic, beauty, grace, brightness and harmony. The article reads well without and, from my perspective as a reviewer, sustains openness to uncertainty and the unknown potential of co-created contact - which is clearly what the authors are seeking. The writing on aesthetic criterion (lines 103-108) could be edited without losing the core power of the diagnostic theme. Similarly on line 37 the word beauty could be omitted and on line 189 the term grace could be omitted. The term aesthetic in the title could be edited from the title.

The idea to skip the term aesthetic from the paper is really a creativity provoking, since it provides a new perspective on the paper and on our work, such a discussion would deserve more time and space. For now, we have edited the paper in order to limit the problematicity of the language.

 The article is definitely publishable on the strength of the discussion about psychopathology, diagnosis and a Gestalt perspective on working with depression. This writing is articulate, accessible and useful for its intelligent, in depth research into the phenomenological process of a therapeutic meeting.

Round 2

Reviewer 2 Report

You have addressed all of my initial concerns and the  revisions have improved the submission.  This is commendable.   

In some of the revisions, however, new errors of written expression, spelling, grammar and so on have been introduced. So in order to be ready for publication the submission requires a thorough edit for written expression and presentation. 

Otherwise, this is a solid contribution that will merit response and dialogue among theoreticians and practitioners of the Gestalt method.